# Mental disorders, COVID-19-related life-saving measures and mortality in France: A nationwide cohort study

Michaël Schwarzinger[1,2]*, Stéphane Luchini[3], Miriam Teschl[3], François Alla[1,2], Vincent Mallet[4,5☉], Jürgen Rehm[6,7,8,9,10,11☉]

1 Department of methodology and innovation in prevention, Bordeaux University Hospital, Bordeaux, France, 2 University of Bordeaux, Inserm UMR 1219-Bordeaux Population Health, Bordeaux, France, 3 Aix-Marseille University, CNRS, EHESS, Centrale Marseille, Aix-Marseille School of Economics, Marseille, France, 4 Université Paris Cité, Paris, France, 5 AP-HP. Centre Université Paris Centre, Groupe Hospitalier Cochin Port Royal, DMU Cancérologie et spécialités médico-chirurgicales, Service d'Hépatologie, Paris, France, 6 Campbell Family Mental Health Research Institute, CAMH, Toronto, Ontario, Canada, 7 Department of Psychiatry, University of Toronto, Toronto, Ontario, Canada, 8 Institute for Mental Health Policy Research, CAMH, Toronto, Ontario, Canada, 9 Dalla Lana School of Public Health, University of Toronto, Toronto, Ontario, Canada, 10 Institute of Medical Science (IMS), University of Toronto, Toronto, Ontario, Canada, 11 Institute for Clinical Psychology and Psychotherapy, TU Dresden, Dresden, Germany

☉ These authors contributed equally to this work.
* michael.schwarzinger@chu-bordeaux.fr

**Data Availability Statement:** Subsets of the National Hospital Discharge (PMSI) database cannot be shared publicly because of legal restrictions on sharing potentially re-identifying

## Abstract

### Background

Meta-analyses have shown that preexisting mental disorders may increase serious Coronavirus Disease 2019 (COVID-19) outcomes, especially mortality. However, most studies were conducted during the first months of the pandemic, were inconclusive for several categories of mental disorders, and not fully controlled for potential confounders. Our study objectives were to assess independent associations between various categories of mental disorders and COVID-19-related mortality in a nationwide sample of COVID-19 inpatients discharged over 18 months and the potential role of salvage therapy triage to explain these associations.

### Methods and findings

We analysed a nationwide retrospective cohort of all adult inpatients discharged with symptomatic COVID-19 between February 24, 2020 and August 28, 2021 in mainland France. The primary exposure was preexisting mental disorders assessed from all discharge information recorded over the last 9 years (dementia, depression, anxiety disorders, schizophrenia, alcohol use disorders, opioid use disorders, Down syndrome, other learning disabilities, and other disorder requiring psychiatric ward admission). The main outcomes were all-cause mortality and access to salvage therapy (intensive-care unit admission or life-saving respiratory support) assessed at 120 days after recorded COVID-19 diagnosis at hospital. Independent associations were analysed in multivariate logistic models.

patient information. According to French laws for secondary analyses of the National Hospital Discharge (PMSI) database (reference methodology MR-005), data are available from the Agence Technique de l'Information Hospitalière (ATIH) (contact via https://www.atih.sante.fr/acces-aux-donnees-pour-les-etablissements-desante-les-chercheurs-et-les-institutionnels ) for researchers who meet all criteria for access to the database.

**Funding:** MS and FA acknowledge funding from the Agence Régionale de Santé de Nouvelle-Aquitaine (ARS-SSMIP). SL and MT acknowledge funding from the French government under the "France 2030" investment plan managed by the French National Research Agency (ANR-17-EURE-0020) and from Excellence Initiative of Aix-Marseille University (A∗MIDEX). JR acknowledges funding from the Institute of Neurosciences, Mental Health and Addiction of the Canadian Institutes of Health Research for the Ontario Canadian Research Initiative Node Team (OCRINT) CRISM Phase II CIHR REN 477887. The funders had no role in study design, data collection and analysis, decision to publish, or preparation of the manuscript.

**Competing interests:** The authors have declared that no competing interests exist.

**Abbreviations:** CI, confidence interval; COVID-19, Coronavirus Disease 2019; OR, odds ratio; SARS-CoV-2, Severe Acute Respiratory Syndrome Coronavirus 2.

Of 465,750 inpatients with symptomatic COVID-19, 153,870 (33.0%) were recorded with a history of mental disorders. Almost all categories of mental disorders were independently associated with higher mortality risks (except opioid use disorders) and lower salvage therapy rates (except opioid use disorders and Down syndrome). After taking into account the mortality risk predicted at baseline from patient vulnerability (including older age and severe somatic comorbidities), excess mortality risks due to caseload surges in hospitals were +5.0% (95% confidence interval (CI), 4.7 to 5.2) in patients without mental disorders (for a predicted risk of 13.3% [95% CI, 13.2 to 13.4] at baseline) and significantly higher in patients with mental disorders (+9.3% [95% CI, 8.9 to 9.8] for a predicted risk of 21.2% [95% CI, 21.0 to 21.4] at baseline). In contrast, salvage therapy rates during caseload surges in hospitals were significantly higher than expected in patients without mental disorders (+4.2% [95% CI, 3.8 to 4.5]) and lower in patients with mental disorders (−4.1% [95% CI, −4.4; −3.7]) for predicted rates similar at baseline (18.8% [95% CI, 18.7-18.9] and 18.0% [95% CI, 17.9-18.2], respectively).

The main limitations of our study point to the assessment of COVID-19-related mortality at 120 days and potential coding bias of medical information recorded in hospital claims data, although the main study findings were consistently reproduced in multiple sensitivity analyses.

## Conclusions

COVID-19 patients with mental disorders had lower odds of accessing salvage therapy, suggesting that life-saving measures at French hospitals were disproportionately denied to patients with mental disorders in this exceptional context.

## Author summary

### Why was this study done?

- Systematic reviews and meta-analyses of previous studies suggest that mental disorders are associated with higher mortality risk in Coronavirus Disease 2019 (COVID-19) patients, but evidence remains limited to the first months of the pandemic, the community setting, and two categories of mental disorders (mood disorders and schizophrenia).

- There is no obvious explanation for the relationship, although the potential role of COVID-19 caseload surges in hospitals that impacted triage decisions for life-saving measures has not been fully explored.

### What did the researchers do and find?

- We examined the associations between various categories of mental disorders and mortality among all inpatients discharged with symptomatic COVID-19 in mainland France, controlling not only for sociodemographic variables and multiple somatic conditions, but also for pandemic periods over 18 months.

- Of 465,750 inpatients discharged with symptomatic COVID-19, one third were recorded with pre-existing mental disorders over the last 9 years, and of 103,890 COVID-19 related deaths, almost half were recorded in patients with pre-existing mental disorders.

- We found independent associations of almost all categories of mental disorders with higher mortality risks and lower salvage therapy rates in COVID-19 inpatients.

- We found that patients with pre-existing mental disorders were disproportionately affected by COVID-19 caseload surges in hospitals with higher-than-expected excess mortality risks and gaps in salvage therapy rates compared to patients without pre-existing mental disorders.

**What do these findings mean?**

- The higher mortality risk for COVID-19 patients with mental disorders raises major ethical issues as it seems this patient group was disproportionately denied life-saving measures at hospital.

- The stability of the study findings should be examined in other jurisdictions.

## Introduction

The Coronavirus Disease 2019 (COVID-19) pandemic may have caused more than 18 million deaths globally in 2020 and 2021 [1]. Risk factors for serious COVID-19 outcomes are older age, male sex, deprivation, and preexisting somatic conditions [2–4]. Preexisting mental disorders also seem to increase the risk of serious COVID-19 outcomes, especially mortality [5,6]. In one meta-analysis based on 11 cross-sectional and longitudinal studies including 204,251 COVID-19 patients, the pooled adjusted odds ratio (OR) of COVID-19-related mortality for any mental disorder was 1.31 (95% confidence interval (CI), 1.13 to 1.52) [5]. Similar results were found in another meta-analysis relying on 16 population-based cohort studies using medico-administrative or electronic health records databases (pooled adjusted OR: 1.38 [95% CI, 1.15 to 1.65]) [6].

However, most studies were conducted during the first wave of the COVID-19 pandemic, with potential bias due to higher COVID-19-related mortality risk [7–9]. In addition, study results were heterogeneous regarding the study setting, scope, and definition of mental disorders [5,6]. Most studies were conducted in the community, and subgroup analyses were inconclusive at hospital [5]. The association found between any mental disorder and COVID-19-related mortality was primarily driven by mood disorders and schizophrenia, while subgroup analyses were inconclusive for other categories of mental disorders [5,6]. Most importantly, there was inconsistent and often insufficient adjustment made for potential confounders [5]. As mental disorders are associated with risk factors for serious COVID-19 outcomes including somatic conditions [10–12], it is not clear if the association found between preexisting mental disorders and COVID-19-related mortality was due to unmeasured confounding [13].

In this study, we aimed at assessing the association between preexisting mental disorders and COVID-19-related mortality, while addressing previous shortcomings with use of the

French National Hospital Discharge database [14]. First, we were able to examine the impact of different COVID-19 wave periods in mainland France up to August 2021 with follow-up until December 2021. Second, since the hospital database comprises all patients discharged with a COVID-19 diagnosis in mainland France, our results are representative for this region, and the sample size was sufficient to detect small effect sizes for various categories of mental disorders. Third, with all hospital data available over the last 9 years for each patient, we were able to adjust the study findings for multiple potential confounders. Finally, there is no obvious explanation for the association of preexisting mental disorders with COVID-19-related mortality that has been attributed to barriers to care, social and lifestyle factors, higher rates of somatic conditions, and biological processes [5,6]. In this study, we assumed that COVID-19 caseload surges in hospitals impacted triage decisions for life-saving measures and we assessed whether COVID-19 patients with preexisting mental disorders had lower chances to access salvage therapy (intensive-care unit admission or life-saving respiratory support) compared to others.

## Methods

### Study design

The data source was the French National Hospital Discharge database (Programme de Médicalisation des Systèmes d'Information [PMSI]), which contains all public and private claims for acute hospital admissions, post-acute, and psychiatric care on a 10-year rolling basis. The standardised discharge summary includes: patient demographics (sex, age at entry, postal code of residency); primary and associated discharge diagnosis codes according to the WHO International Classification of Diseases, 10th revision, French version (ICD-10-FR); medical procedures performed; entry and discharge dates and modes (including in-hospital death). Using unique anonymous identifiers based on encrypted Social Security numbers, the hospital trajectory of each selected patient can be traced over time [14,15].

We included all adult patients aged 18 years and older, residing in mainland France, who were discharged from acute hospitals with a COVID-19 diagnosis record (ICD-10-FR: U07.10, U07.11, U.07.12, U07.14, U07.15, U10.9) between February 24, 2020 and August 28, 2021. We excluded all patients discharged alive after day-case admission (i.e., at lower mortality risk) or recorded with asymptomatic Severe Acute Respiratory Syndrome Coronavirus 2 (SARS-CoV-2) infection (U07.12) (i.e., at mortality risk unrelated to COVID-19). The full coding dictionary of the study is provided with supporting references in **Appendix A in S1 Text**.

Data management and secondary analyses of the National Hospital Discharge database were performed in accordance with French laws for this type of research studies (reference methodology MR-005) [16]. Accordingly, the protocol of this study was submitted to the Health Data Hub (registration number F20220215185433 delivered on 02/15/2022, available in French language at https://www.health-data-hub.fr/projets/fard-oh). The approval of an Institutional Review Board was not required because hospital discharge data are fully anonymous. For the same reason, informed consent is not possible and not required. The study complies with the RECORD statement (**Appendix B in S1 Text**) [17].

### Procedures

To limit ascertainment bias of COVID-19-related mortality at first acute hospital discharge, all patients were followed over 120 days after the first date of recorded COVID-19 diagnosis (i.e., patients first recorded with COVID-19 on August 28, 2021 were followed until December 27, 2021), and all-cause mortality was assessed from all acute, post-acute, and psychiatric hospitals

[18]. To take into account life-saving respiratory support that may have been performed outside intensive-care units and evolving medical procedures, we assessed salvage therapy by the first record of intensive-care unit admission, extracorporeal membrane oxygenation, invasive mechanical respiratory support, or continuous positive airway pressure therapy over the follow-up period [19].

To assess the effects of COVID-19 caseload surges in hospitals on individual prognosis [7–9], we defined wave periods according to official epidemiological reports [20] and the admission of at least 3,500 new inpatients per week in the study sample. Accordingly, we considered 4 wave periods (first: weeks 11 to 19 of 2020; second: weeks 37 of 2020 to 8 of 2021; third: weeks 9 to 19 in 2021; fourth: weeks 31 to 34 in 2021) and 2 inter-wave periods (first: weeks 20 to 36 of 2020; second: weeks 20 to 30 of 2021).

Sociodemographic prognostic factors included [2–4]: sex, age assessed in 5-year categories (less than 30; 30 to 34;. . . 85 to 89; 90 or more), area deprivation index quintile with use of a validated index computed for the 5,645 postal codes of residency in mainland France (FDep 2015) [21], and 5 main French regions as Ile-de-France and North-East regions were more severely hit during the first wave of the pandemic [22].

Other individual prognostic factors were assessed at the first date of recorded COVID-19 diagnosis at hospital with use of all discharge information recorded in all French hospitals over the last 9 years. We identified preexisting mental disorders based on previous meta-analyses [5,6] with use of 8 non-mutually exclusive categories (dementia, depression, anxiety disorders, schizophrenia, alcohol use disorders, opioid use disorders, Down syndrome, and other learning disabilities) and other disorder requiring psychiatric ward admission.

Risk factors identified early on for severe COVID-19 (i.e., acute hospital admission) were tobacco smoking, obesity, hypertension, and diabetes [23]. Severe somatic conditions were assessed with the Charlson Comorbidity Index considering only comorbidities that were independently associated with prognostic outcome in the French National Hospital Discharge database (congestive heart failure, peripheral vascular disease, cerebrovascular disease, chronic pulmonary disease, hemiplegia, liver disease, renal disease, solid tumour, haematological malignancy, AIDS) [24]. Transplant recipients were also identified [2,3]. To limit residual confounding from multiple other severe somatic conditions [2–4], we assessed the delay between the latest acute hospital discharge for any reason other than pregnancy and psychiatry and first COVID-19 diagnosis record: SARS-CoV-2 infection during hospital care (i.e., first COVID-19 diagnosis recorded after hospital entry for another reason), previous discharge in the last 3 months, 4 to 12 months, 2 to 3 years, or 4 to 9 years, and no previous admission in the last 9 years.

## Statistical procedures

The independent associations of each category of mental disorder with all-cause mortality or access to salvage therapy were estimated in 2 multivariate logistic models adjusting for all prognostic factors without variable selection. The robustness of associations with all-cause mortality was further assessed considering salvage therapy simultaneously. In a first sensitivity analysis, we assessed independent associations with all-cause mortality depending on access to salvage therapy using a simultaneous probit multivariate model with 3 dependent variables (access to salvage therapy, mortality with salvage therapy, and mortality without salvage therapy) and joint estimation to control for unobserved heterogeneity and omitted variable bias [25,26]. In response to the peer review process, we conducted a second sensitivity analysis to assess the direct effect of each category of mental disorder on all-cause mortality that is not due to mediation or interaction with salvage therapy (so-called "controlled direct effect")

using a series of causal mediation analyses adjusting for all other prognostic factors and assuming no unobserved confounding factors [27,28].

In a second set of analyses, we aimed at further disentangling the effects of patient vulnerability and COVID-19 caseload surges in hospitals on the outcomes of COVID-19 inpatients with preexisting mental disorders. Based on the first set of analyses, we first considered patients without mental disorders admitted during the first inter-wave period as a reference group with the best prognostic outcomes. Then, we relied on this reference group to predict the mortality risk expected in each patient group defined by mental disorder status and pandemic period with use of a multivariate logistic model (after excluding preexisting mental disorder and period covariates) [29–31]. The comparison of mortality risks predicted at baseline across patient groups allows assessing the effect on all-cause mortality of patient case-mix and latent vulnerability level as defined by all covariates other than preexisting mental disorder and pandemic period. Finally, we computed the excess mortality risk in each patient group as the difference between observed and predicted mortality risks. The comparison of excess mortality risks between patients with and without mental disorders by pandemic period allows assessing the effect of COVID-19 caseload surges in hospitals in each patient group, irrespective of the mortality risk predicted at baseline. This approach was similarly followed to assess potential gaps in salvage therapy.

To ascertain excess mortality risks and gaps in salvage therapy, we did several subgroup analyses (18 to 64 or 65 years and above; each category of mental disorder) and sensitivity analyses depending on COVID-19 case definition (selection of inpatients with COVID-19 respiratory symptoms or admitted for symptomatic COVID-19 [i.e., exclusion of inpatients with SARS-CoV-2 infection during hospital care]), salvage therapy definition (selection of admissions to intensive-care units), or censoring date (28 days after the date of recorded COVID-19 diagnosis at hospital or at first acute hospital discharge including all inter-hospital transfers during the same course of acute hospital care).

All analyses were done with SAS (version 9.4) including PROC QLIM (simultaneous probit multivariate model) and PROC CAUSALMED (causal mediation analyses).

## Results

Between February 24, 2020 and August 28, 2021, 558,323 adults residing in mainland France were discharged with a COVID-19 diagnosis record from all acute hospitals. After excluding 52,009 (9.3%) patients discharged alive after day-case admission and 40,564 (7.3%) recorded with asymptomatic SARS-CoV-2 infection, 465,750 inpatients with symptomatic COVID-19 were followed in the study cohort.

The burden of the COVID-19 pandemic on French acute hospitals was marked by 4 wave periods (**Fig 1**), while patient characteristics varied across periods (**Table A in S1 Text**). Median (interquartile range) age increased from the first (72 [58 to 84] years) to the second wave period (75 [62 to 85] years) in 2020 and then decreased in 2021, potentially in relation to COVID-19 vaccination uptake (third and fourth wave periods: 68 [55 to 80] and 62 [47 to 77] years, respectively) ($P < 0.001$). Similarly, the bulk of patients recorded with histories of preexisting mental disorders (33.0%), risk factors for severe COVID-19 (72.4%), severe somatic conditions (57.1%), and acute hospital admission (83.8%) decreased in 2021 (all $P < 0.001$).

A total of 103,890 (22.3%) inpatients died within 120 days after the first date of COVID-19 diagnosis record (**Table 1**). All other things being equal, the first inter-wave period was associated with the lowest mortality risk compared to other pandemic periods ($P < 0.001$). Age was strongly associated with higher mortality risk ($P < 0.001$). Except for opioid use disorders,

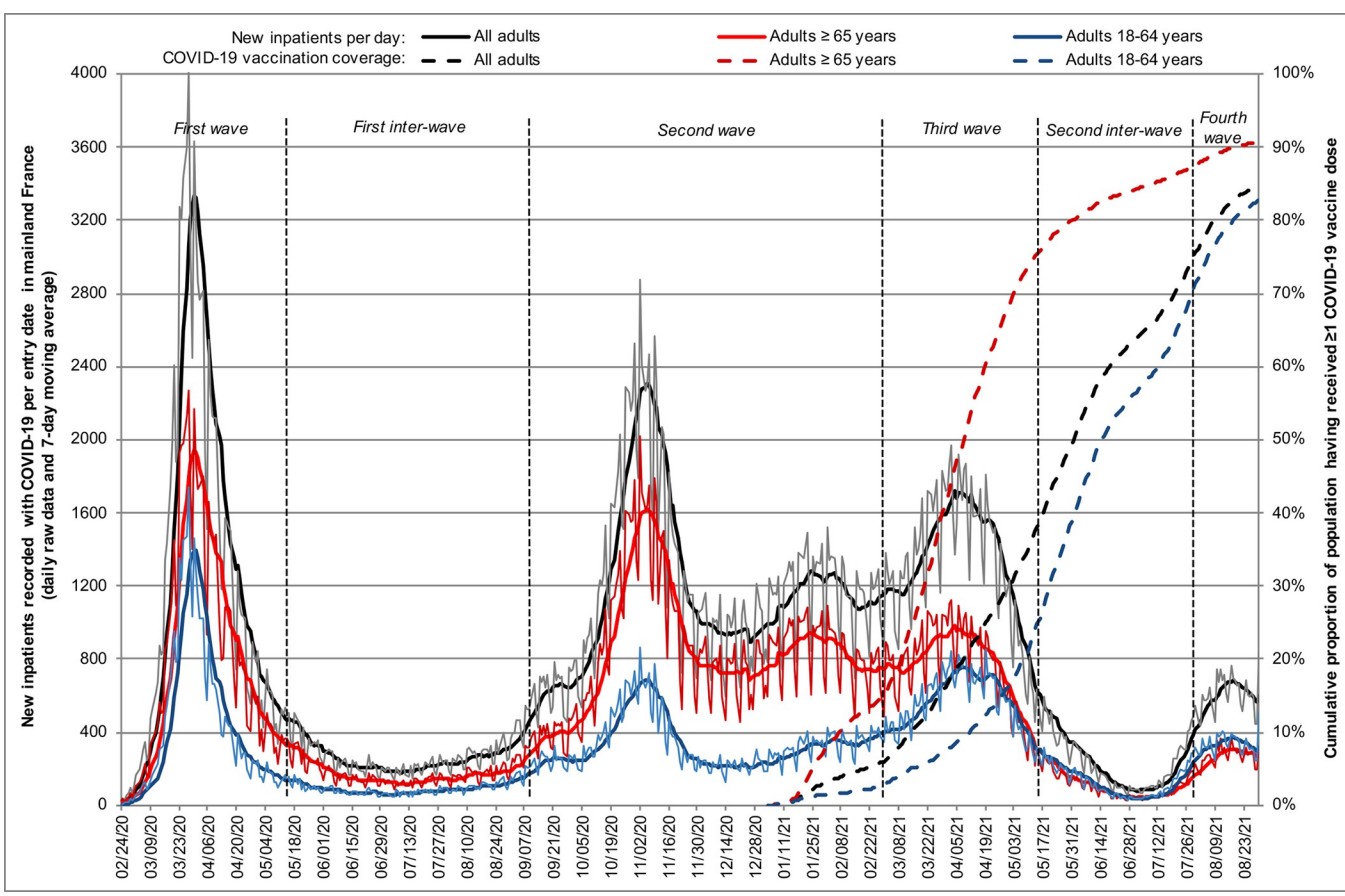

**Fig 1. Burden of COVID-19 pandemic on French acute hospitals over 18 months (*n* = 465,750).**

each category of mental disorder was independently associated with higher mortality risk (all *P* < 0.05).

A total of 92,986 (20.0%) inpatients had access to salvage therapy (**Table 2**). All other things being equal, the rate of salvage therapy increased over the study period (*P* < 0.001). Age was strongly associated with salvage therapy in an inverted-U shaped relationship peaking at 65 to 69 years (*P* < 0.001). Except for opioid use disorders and Down syndrome, each category of mental disorder was independently associated with a lower rate of salvage therapy (all *P* < 0.01).

Independent associations of preexisting mental disorders with higher mortality risk were similarly found in 2 sensitivity analyses considering salvage therapy simultaneously. Except for opioid use disorders, most categories of mental disorders were independently associated with higher mortality risk in patients with or without salvage therapy in a simultaneous multivariate probit model (**Tables B and C in S1 Text**). Except for opioid use disorders, each category of mental disorders had a direct effect on higher mortality risk that is not due to mediation or interaction with salvage therapy in a causal mediation analysis (**Table D in S1 Text**). However, both sensitivity analyses pointed to unobserved confounding factors associated with a higher access to salvage therapy in patients with better prognosis, all other observed factors being equal.

COVID-19 caseload surges in hospitals were independently associated with higher mortality risks (**Table 1**). Compared to the first inter-wave period, patients without mental disorders had a significantly worse prognosis than expected in all other periods (**Fig 2**, comparison of

**Table 1. Risk factors for 120-day mortality of inpatients with symptomatic COVID-19 in France.**

| Risk factors | All patients (%) 465,750 (100.0) | 120-day mortality (%) 103,890 (22.3) | Adjusted OR (95% CI) | *p*-value |
|---|---|---|---|---|
| COVID-19 pandemic period | | | | |
| First wave (2020 weeks 11–19) | 99,633 (21.4) | 23,006 (23.1) | 1.29 (1.24–1.33) | <0.001 |
| First inter-wave (2020 weeks 20–36) | 31,840 (6.8) | 5,873 (18.4) | 1 | |
| Second wave (2020 week 37–2021 week 8) | 202,771 (43.5) | 51,382 (25.3) | 1.34 (1.29–1.38) | |
| Third wave (2021 weeks 9–19) | 99,430 (21.3) | 18,876 (19.0) | 1.33 (1.28–1.38) | |
| Second inter-wave (2021 weeks 20–30) | 17,352 (3.7) | 2,423 (14.0) | 1.19 (1.12–1.26) | |
| Fourth wave (2021 weeks 31–34) | 14,724 (3.2) | 2,330 (15.8) | 1.44 (1.35–1.53) | |
| COVID-19 with respiratory symptoms | 371,016 (79.7) | 87,114 (23.5) | 2.55 (2.49–2.61) | <0.001 |
| COVID-19-related multisystem inflammatory syndrome | 110,785 (23.8) | 27,599 (24.9) | 1.70 (1.66–1.73) | <0.001 |
| Male | 251,360 (54.0) | 60,080 (23.9) | 1.44 (1.42–1.47) | <0.001 |
| Age, median (IQR) years | 72 (58–84) | 82 (73–89) | | |
| ≥90 years | 50,512 (10.8) | 22,261 (44.1) | 53.23 (44.93–63.07) | <0.001 |
| 85–89 years | 56,848 (12.2) | 21,819 (38.4) | 38.27 (32.30–45.33) | |
| 80–84 years | 52,601 (11.3) | 17,576 (33.4) | 28.73 (24.26–34.03) | |
| 75–79 years | 45,120 (9.7) | 12,302 (27.3) | 19.76 (16.68–23.41) | |
| 70–74 years | 52,002 (11.2) | 11,348 (21.8) | 14.32 (12.09–16.97) | |
| 65–69 years | 44,076 (9.5) | 7,703 (17.5) | 10.80 (9.11–12.80) | |
| 60–64 years | 37,901 (8.1) | 4,729 (12.5) | 7.58 (6.39–8.99) | |
| 55–59 years | 33,255 (7.1) | 2,809 (8.4) | 5.30 (4.46–6.29) | |
| 50–54 years | 26,336 (5.7) | 1,593 (6.0) | 3.90 (3.27–4.65) | |
| 45–49 years | 19,896 (4.3) | 826 (4.2) | 2.79 (2.33–3.35) | |
| 40–44 years | 13,792 (3.0) | 404 (2.9) | 2.07 (1.70–2.52) | |
| 35–39 years | 11,438 (2.5) | 241 (2.1) | 1.58 (1.28–1.95) | |
| 30–34 years | 9,508 (2.0) | 139 (1.5) | 1.19 (0.93–1.50) | |
| 18–29 years | 12,465 (2.7) | 140 (1.1) | 1 | |
| Area deprivation index quintile | | | | |
| FDeP 5 (most deprived) | 114,480 (24.6) | 26,279 (23.0) | 1.23 (1.20–1.26) | <0.001 |
| FDeP 4 | 92,877 (19.9) | 21,329 (23.0) | 1.17 (1.14–1.20) | |
| FDeP 3 | 85,390 (18.3) | 19,012 (22.3) | 1.15 (1.12–1.19) | |
| FDep 2 | 84,253 (18.1) | 18,175 (21.6) | 1.13 (1.10–1.16) | |
| FDep 1 (least deprived) | 88,750 (19.1) | 19,095 (21.5) | 1 | |
| Residency in mainland France | | | | |
| North-East region | 121,912 (26.2) | 30,090 (24.7) | 0.97 (0.95–1.00) | <0.001 |
| North-West region | 74,331 (16.0) | 16,201 (21.8) | 0.81 (0.79–0.83) | |
| South-East region | 126,423 (27.1) | 27,659 (21.9) | 0.85 (0.83–0.87) | |
| South-West region | 34,205 (7.3) | 7,190 (21.0) | 0.80 (0.77–0.83) | |
| Ile-de-France region | 108,879 (23.4) | 22,750 (20.9) | 1 | |
| Any preexisting mental disorder | 153,870 (33.0) | 46,983 (30.5) | | |
| Dementia | 67,539 (14.5) | 24,034 (35.6) | 1.03 (1.01–1.05) | 0.014 |
| Depression | 49,420 (10.6) | 15,273 (30.9) | 1.07 (1.04–1.10) | <0.001 |
| Anxiety disorders | 46,039 (9.9) | 14,611 (31.7) | 1.04 (1.01–1.07) | 0.018 |
| Schizophrenia | 13,400 (2.9) | 3,149 (23.5) | 1.15 (1.10–1.21) | <0.001 |
| Alcohol use disorders | 36,509 (7.8) | 9,695 (26.6) | 1.15 (1.12–1.19) | <0.001 |
| Opioid use disorders | 3,088 (0.7) | 547 (17.7) | 0.96 (0.87–1.07) | 0.50 |
| Down syndrome | 1,122 (0.2) | 220 (19.6) | 4.44 (3.79–5.20) | <0.001 |
| Other learning disabilities | 5,077 (1.1) | 948 (18.7) | 1.74 (1.61–1.88) | <0.001 |

*(Continued)*

**Table 1.**  (Continued)

| Risk factors | All patients (%) | 120-day mortality (%) | Adjusted OR (95% CI) | p-value |
|---|---|---|---|---|
| | 465,750 (100.0) | 103,890 (22.3) | | |
| Other disorder with psychiatric ward admission | 2,869 (0.6) | 958 (33.4) | 1.26 (1.15–1.38) | <0.001 |
| Any risk factor for severe COVID-19 | 337,312 (72.4) | 84,759 (25.1) | | |
| Tobacco smoking | 57,209 (12.3) | 13,438 (23.5) | 1.00 (0.97–1.02) | 0.77 |
| Obesity (BMI $\geq$ 30 kg/m$^2$) | 117,385 (25.2) | 23,578 (20.1) | 0.93 (0.91–0.95) | <0.001 |
| Hypertension | 270,818 (58.1) | 75,358 (27.8) | 0.83 (0.81–0.85) | <0.001 |
| Diabetes mellitus | 135,162 (29.0) | 34,491 (25.5) | 1.03 (1.01–1.05) | <0.001 |
| Any severe somatic comorbidity | 265,912 (57.1) | 82,964 (31.2) | | |
| Congestive heart failure | 114,419 (24.6) | 43,049 (37.6) | 1.39 (1.36–1.41) | <0.001 |
| Peripheral vascular disease | 54,756 (11.8) | 19,723 (36.0) | 1.12 (1.09–1.14) | <0.001 |
| Cerebrovascular disease | 65,265 (14.0) | 23,017 (35.3) | 1.11 (1.09–1.14) | <0.001 |
| Chronic pulmonary disease | 81,971 (17.6) | 23,006 (28.1) | 1.03 (1.01–1.05) | 0.013 |
| Hemiplegia | 34,189 (7.3) | 11,366 (33.2) | 1.29 (1.25–1.33) | <0.001 |
| Moderate or severe liver disease | 7,827 (1.7) | 3,046 (38.9) | 2.45 (2.32–2.59) | <0.001 |
| Mild liver disease | 19,077 (4.1) | 4,515 (23.7) | 1.17 (1.12–1.21) | |
| Moderate or severe renal disease | 69,394 (14.9) | 27,043 (39.0) | 1.30 (1.27–1.33) | <0.001 |
| Metastatic solid tumour | 26,008 (5.6) | 13,231 (50.9) | 4.55 (4.42–4.69) | <0.001 |
| Solid tumour without metastasis | 43,065 (9.2) | 13,663 (31.7) | 1.25 (1.22–1.28) | |
| Haematological malignancy | 15,226 (3.3) | 6,044 (39.7) | 1.80 (1.73–1.87) | <0.001 |
| AIDS | 1,802 (0.4) | 265 (14.7) | 1.05 (0.91–1.22) | 0.48 |
| Transplant recipient | 8,397 (1.8) | 2,109 (25.1) | 1.19 (1.12–1.26) | <0.001 |
| Delay between the latest acute hospital discharge and first COVID-19 record | | | | |
| SARS-CoV-2 infection during hospital care | 70,427 (15.1) | 25,714 (36.5) | 1.97 (1.90–2.03) | <0.001 |
| Previous discharge in the last 3 months | 83,000 (17.8) | 25,530 (30.8) | 1.51 (1.46–1.56) | |
| Previous discharge in the last 4–12 months | 92,456 (19.9) | 18,363 (19.9) | 1.11 (1.08–1.15) | |
| Previous discharge in the last 2–3 years | 71,909 (15.4) | 15,891 (22.1) | 1.12 (1.08–1.15) | |
| Previous discharge in the last 4–9 years | 72,582 (15.6) | 11,752 (16.2) | 1.10 (1.06–1.14) | |
| No previous admission in the last 9 years | 75,376 (16.2) | 6,640 (8.8) | 1 | |

Adjusted ORs and 95% CIs were estimated in multivariate logistic models without variable selection. For binary variables, the reference category is the absence of hospital record (COVID-19 characteristics; each category of mental disorder, risk factors for severe COVID-19, or severe somatic comorbidities).

AIDS, acquired immunodeficiency syndrome; BMI body mass index; CI, confidence interval; COVID-19, Coronavirus Disease 2019; IQR, interquartile range; OR, odds ratio; SARS-CoV-2, Severe Acute Respiratory Syndrome Coronavirus 2.

first and second blue bars). In patients without mental disorders, 15,513 (95% CI, 14,679 to 16,347) of 56,907 COVID-19-related deaths may have been avoided overall without COVID-19 caseload surges in hospitals (excess mortality risk of +5.0% [95% CI, 4.7 to 5.2] for a predicted risk of 13.3% [95% CI, 13.2 to 13.4] at baseline).

Patients with preexisting mental disorders were recorded with more prognostic factors for COVID-19-related mortality than others (**Tables E and F in S1 Text**) that involved a worse mortality risk predicted at baseline in all periods (**Fig 2**, comparison of first blue and first red bars). COVID-19 caseload surges in hospitals were even more negatively associated with individual prognosis in patients with mental disorders compared to others (**Fig 2**, comparison of second blue and second red bars). In patients with mental disorders, 14,343 (95% CI, 13,622 to 15,064) of 46,983 COVID-19-related deaths may have been avoided overall without COVID-19 caseload surges in hospitals (excess mortality risk of +9.3% [95% CI, 8.9 to 9.8] for a predicted risk of 21.2% [95% CI, 21.0 to 21.4] at baseline).

**Table 2. Risk factors for salvage therapy of inpatients with symptomatic COVID-19 in France.**

| Risk factors | All patients (%) | Salvage therapy (%) | Adjusted OR (95% CI) | p-value |
|---|---|---|---|---|
| | 465,750 (100.0) | 92,986 (20.0) | | |
| COVID-19 pandemic period | | | | |
| First wave (2020 weeks 11–19) | 99,633 (21.4) | 19,276 (19.3) | 0.93 (0.90–0.97) | <0.001 |
| First inter-wave (2020 weeks 20–36) | 31,840 (6.8) | 5,006 (15.7) | 1 | |
| Second wave (2020 week 37–2021 week 8) | 202,771 (43.5) | 37,890 (18.7) | 1.00 (0.96–1.03) | |
| Third wave (2021 weeks 9–19) | 99,430 (21.3) | 23,278 (23.4) | 1.13 (1.09–1.17) | |
| Second inter-wave (2021 weeks 20–30) | 17,352 (3.7) | 3,835 (22.1) | 1.17 (1.11–1.23) | |
| Fourth wave (2021 weeks 31–34) | 14,724 (3.2) | 3,701 (25.1) | 1.37 (1.30–1.45) | |
| COVID-19 with respiratory symptoms | 371,016 (79.7) | 82,792 (22.3) | 3.46 (3.36–3.55) | <0.001 |
| COVID-19-related multisystem inflammatory syndrome | 110,785 (23.8) | 25,930 (23.4) | 2.25 (2.20–2.29) | <0.001 |
| Male | 251,360 (54.0) | 61,029 (24.3) | 1.50 (1.48–1.53) | <0.001 |
| Age, median (IQR) years | 72 (58–84) | 67 (56–74) | | |
| ≥90 years | 50,512 (10.8) | 1,787 (3.5) | 0.23 (0.21–0.25) | <0.001 |
| 85–89 years | 56,848 (12.2) | 3,900 (6.9) | 0.42 (0.40–0.45) | |
| 80–84 years | 52,601 (11.3) | 6,649 (12.6) | 0.77 (0.73–0.82) | |
| 75–79 years | 45,120 (9.7) | 10,555 (23.4) | 1.47 (1.38–1.56) | |
| 70–74 years | 52,002 (11.2) | 15,172 (29.2) | 1.88 (1.77–1.99) | |
| 65–69 years | 44,076 (9.5) | 14,026 (31.8) | 2.07 (1.96–2.20) | |
| 60–64 years | 37,901 (8.1) | 11,609 (30.6) | 1.96 (1.85–2.07) | |
| 55–59 years | 33,255 (7.1) | 9,265 (27.9) | 1.72 (1.62–1.82) | |
| 50–54 years | 26,336 (5.7) | 6,948 (26.4) | 1.60 (1.51–1.70) | |
| 45–49 years | 19,896 (4.3) | 4,623 (23.2) | 1.38 (1.29–1.46) | |
| 40–44 years | 13,792 (3.0) | 3,004 (21.8) | 1.30 (1.21–1.39) | |
| 35–39 years | 11,438 (2.5) | 2,196 (19.2) | 1.18 (1.10–1.26) | |
| 30–34 years | 9,508 (2.0) | 1,503 (15.8) | 1.00 (0.93–1.08) | |
| 18–29 years | 12,465 (2.7) | 1,749 (14.0) | 1 | |
| Area deprivation index quintile | | | | |
| FDeP 5 (most deprived) | 114,480 (24.6) | 23,345 (20.4) | 1.02 (1.00–1.05) | <0.001 |
| FDeP 4 | 92,877 (19.9) | 17,425 (18.8) | 0.97 (0.94–0.99) | |
| FDeP 3 | 85,390 (18.3) | 17,072 (20.0) | 1.04 (1.01–1.07) | |
| FDep 2 | 84,253 (18.1) | 17,309 (20.5) | 1.02 (0.99–1.05) | |
| FDep 1 (least deprived) | 88,750 (19.1) | 17,835 (20.1) | 1 | |
| Residency in mainland France | | | | |
| North-East region | 121,912 (26.2) | 23,473 (19.3) | 0.84 (0.82–0.86) | <0.001 |
| North-West region | 74,331 (16.0) | 12,205 (16.4) | 0.76 (0.74–0.78) | |
| South-East region | 126,423 (27.1) | 25,112 (19.9) | 0.91 (0.89–0.94) | |
| South-West region | 34,205 (7.3) | 6,942 (20.3) | 1.02 (0.99–1.06) | |
| Ile-de-France region | 108,879 (23.4) | 25,254 (23.2) | 1 | |
| Any preexisting mental disorder | 153,870 (33.0) | 21,500 (14.0) | | |
| Dementia | 67,539 (14.5) | 3,836 (5.7) | 0.41 (0.39–0.42) | <0.001 |
| Depression | 49,420 (10.6) | 6,110 (12.4) | 0.83 (0.80–0.86) | <0.001 |
| Anxiety disorders | 46,039 (9.9) | 5,342 (11.6) | 0.85 (0.82–0.88) | <0.001 |
| Schizophrenia | 13,400 (2.9) | 2,409 (18.0) | 0.93 (0.89–0.98) | 0.004 |
| Alcohol use disorders | 36,509 (7.8) | 8,681 (23.8) | 0.89 (0.86–0.92) | <0.001 |
| Opioid use disorders | 3,088 (0.7) | 698 (22.6) | 1.11 (1.01–1.22) | 0.026 |
| Down syndrome | 1,122 (0.2) | 242 (21.6) | 1.00 (0.86–1.15) | 0.97 |
| Other learning disabilities | 5,077 (1.1) | 916 (18.0) | 0.74 (0.68–0.80) | <0.001 |
| Other disorder with psychiatric ward admission | 2,869 (0.6) | 439 (15.3) | 0.80 (0.72–0.89) | <0.001 |

*(Continued)*

**Table 2.** (Continued)

| Risk factors | All patients (%) | Salvage therapy (%) | Adjusted OR (95% CI) | *p*-value |
|---|---|---|---|---|
| | 465,750 (100.0) | 92,986 (20.0) | | |
| Any risk factor for severe COVID-19 | 337,312 (72.4) | 72,338 (21.4) | | |
| Tobacco smoking | 57,209 (12.3) | 15,458 (27.0) | 1.20 (1.17–1.23) | <0.001 |
| Obesity (BMI $\geq$ 30 kg/m$^2$) | 117,385 (25.2) | 33,394 (28.4) | 1.56 (1.54–1.59) | <0.001 |
| Hypertension | 270,818 (58.1) | 54,874 (20.3) | 1.33 (1.31–1.36) | <0.001 |
| Diabetes mellitus | 135,162 (29.0) | 31,968 (23.7) | 1.07 (1.05–1.09) | <0.001 |
| Any severe somatic comorbidity | 265,912 (57.1) | 51,283 (19.3) | | |
| Congestive heart failure | 114,419 (24.6) | 20,774 (18.2) | 1.18 (1.15–1.20) | <0.001 |
| Peripheral vascular disease | 54,756 (11.8) | 10,599 (19.4) | 0.90 (0.88–0.92) | <0.001 |
| Cerebrovascular disease | 65,265 (14.0) | 9,883 (15.1) | 0.86 (0.83–0.88) | <0.001 |
| Chronic pulmonary disease | 81,971 (17.6) | 17,253 (21.0) | 0.99 (0.97–1.01) | 0.38 |
| Hemiplegia | 34,189 (7.3) | 6,425 (18.8) | 1.05 (1.01–1.08) | 0.011 |
| Moderate or severe liver disease | 7,827 (1.7) | 1,990 (25.4) | 1.07 (1.01–1.13) | 0.042 |
| Mild liver disease | 19,077 (4.1) | 4,683 (24.5) | 0.98 (0.95–1.02) | |
| Moderate or severe renal disease | 69,394 (14.9) | 11,799 (17.0) | 0.95 (0.93–0.98) | <0.001 |
| Metastatic solid tumour | 26,008 (5.6) | 3,886 (14.9) | 0.58 (0.56–0.60) | <0.001 |
| Solid tumour without metastasis | 43,065 (9.2) | 8,012 (18.6) | 0.92 (0.89–0.94) | |
| Haematological malignancy | 15,226 (3.3) | 3,471 (22.8) | 1.24 (1.19–1.30) | <0.001 |
| AIDS | 1,802 (0.4) | 505 (28.0) | 1.08 (0.97–1.21) | 0.16 |
| Transplant recipient | 8,397 (1.8) | 2,540 (30.2) | 1.32 (1.25–1.39) | <0.001 |
| Delay between the latest acute hospital discharge and first COVID-19 record | | | | |
| SARS-CoV-2 infection during hospital care | 70,427 (15.1) | 16,894 (24.0) | 1.56 (1.51–1.60) | <0.001 |
| Previous discharge in the last 3 months | 83,000 (17.8) | 13,089 (15.8) | 0.73 (0.70–0.75) | |
| Previous discharge in the last 4–12 months | 92,456 (19.9) | 17,239 (18.6) | 0.83 (0.81–0.85) | |
| Previous discharge in the last 2–3 years | 71,909 (15.4) | 12,539 (17.4) | 0.80 (0.78–0.82) | |
| Previous discharge in the last 4–9 years | 72,582 (15.6) | 15,248 (21.0) | 0.90 (0.88–0.93) | |
| No previous admission in the last 9 years | 75,376 (16.2) | 17,977 (23.8) | 1 | |

Adjusted ORs and 95% CIs were estimated in multivariate logistic models without variable selection. For binary variables, the reference category is the absence of hospital record (COVID-19 characteristics; each category of mental disorder, risk factors for severe COVID-19, or severe somatic comorbidities).

AIDS, acquired immunodeficiency syndrome; BMI, body mass index; CI, confidence interval; COVID-19, Coronavirus Disease 2019; IQR, interquartile range; OR, odds ratio; SARS-CoV-2, Severe Acute Respiratory Syndrome Coronavirus 2.

In subgroup analyses on age category, differences in excess mortality risks between patients with and without mental disorders were more marked in patients aged 18 to 64 years (42.3% alcohol use disorders) (+5.5% [95% CI, 4.7 to 6.3] versus +1.1% [95% CI, 0.8 to 1.3] for predicted risks of 8.3% [95% CI, 7.9 to 8.6] and 3.8% [95% CI, 3.7 to 3.9] at baseline, respectively) compared to patients older than 65 years (54.4% dementia) (+10.1% [95% CI, 9.6 to 10.7] versus +8.0% [95% CI, 7.5 to 8.4] for predicted risks of 24.9% [95% CI, 24.7 to 25.1] and 20.1% [95% CI, 20.0 to 20.3] at baseline, respectively) (**Figs A and B in S1 Text**).

A similar analysis was carried on access to salvage therapy (**Fig 3**). Patients without mental disorders accessed salvage therapy at significantly higher rates than expected, with increasing rates over the study period (overall: +4.2% [95% CI, 3.8 to 4.5] for a predicted rate of 18.8% [95% CI, 18.7 to 18.9] at baseline). In contrast, patients with preexisting mental disorders accessed salvage therapy at significantly lower rates than expected, except in the last 2 pandemic periods (overall: −4.1% [95% CI, −4.4; −3.7]) for a predicted rate of 18.0% [95% CI, 17.9 to 18.2] at baseline).

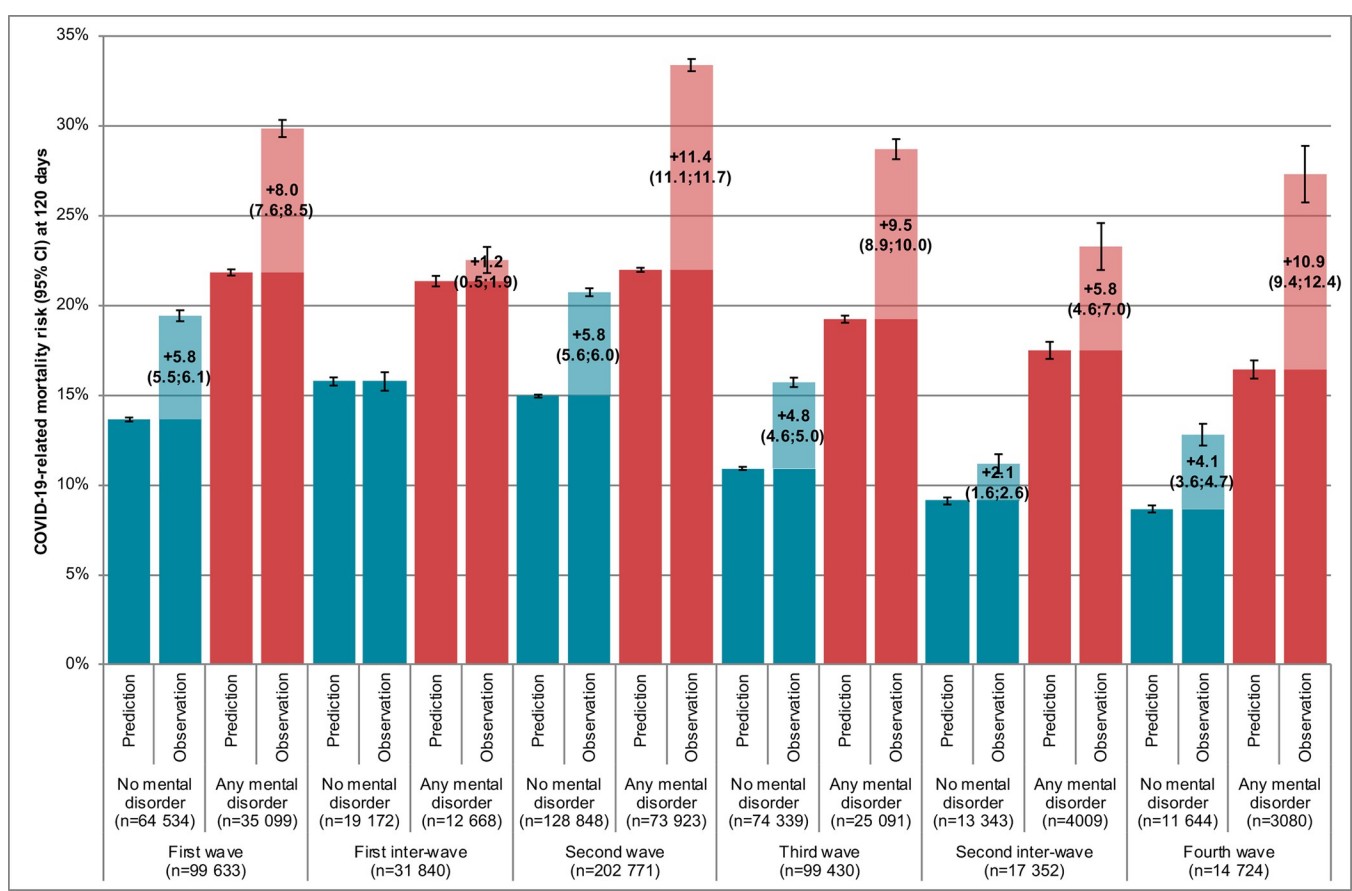

**Fig 2. Associations of pandemic periods and preexisting mental disorders with 120-day mortality among inpatients with symptomatic COVID-19 in France ($n$ = 465,750).**

In subgroup analyses on age category, differences in salvage therapy rates between patients with and without mental disorders were more marked in patients older than 65 years (−5.7% [95% CI, −6.0; −5.3] versus +3.4% [95% CI, 3.0 to 3.8] for predicted rates of 16.3% [95% CI, 16.1 to 16.4] and 18.4% [95% CI, 18.3 to 18.5] at baseline, respectively) compared to patients aged 18 to 64 years (+1.2% [95% CI, 0.1 to 2.3] versus +5.2% [95% CI, 4.7 to 5.7] for predicted rates of 25.2% [95% CI, 24.8 to 25.6] and 19.2% [95% CI, 19.0 to 19.4] at baseline, respectively) (**Figs C and D in S1 Text**).

Differences in excess mortality risks and gaps in salvage therapy rates were corroborated for each category of mental disorder (**Table G in S1 Text**). Overall, the study findings were consistently reproduced in sensitivity analyses selecting 371,016 (79.7%) inpatients with COVID-19 respiratory symptoms (**Figs E and F in S1 Text**) or 395,323 (84.9%) inpatients admitted for symptomatic COVID-19 (**Figs G and H in S1 Text**), considering only intensive-care unit admissions (81,686 [17.5%] inpatients; **Fig I in S1 Text**), or censoring follow-up at 28 days (80,174 [17.2%] deaths; **Fig J in S1 Text**) or first acute hospital discharge (89,620 [19.2%] deaths; **Fig K in S1 Text**).

## Discussion

We reported that in a nationwide cohort study of all 465,750 inpatients discharged with symptomatic COVID-19 in mainland France, preexisting mental disorders were associated with

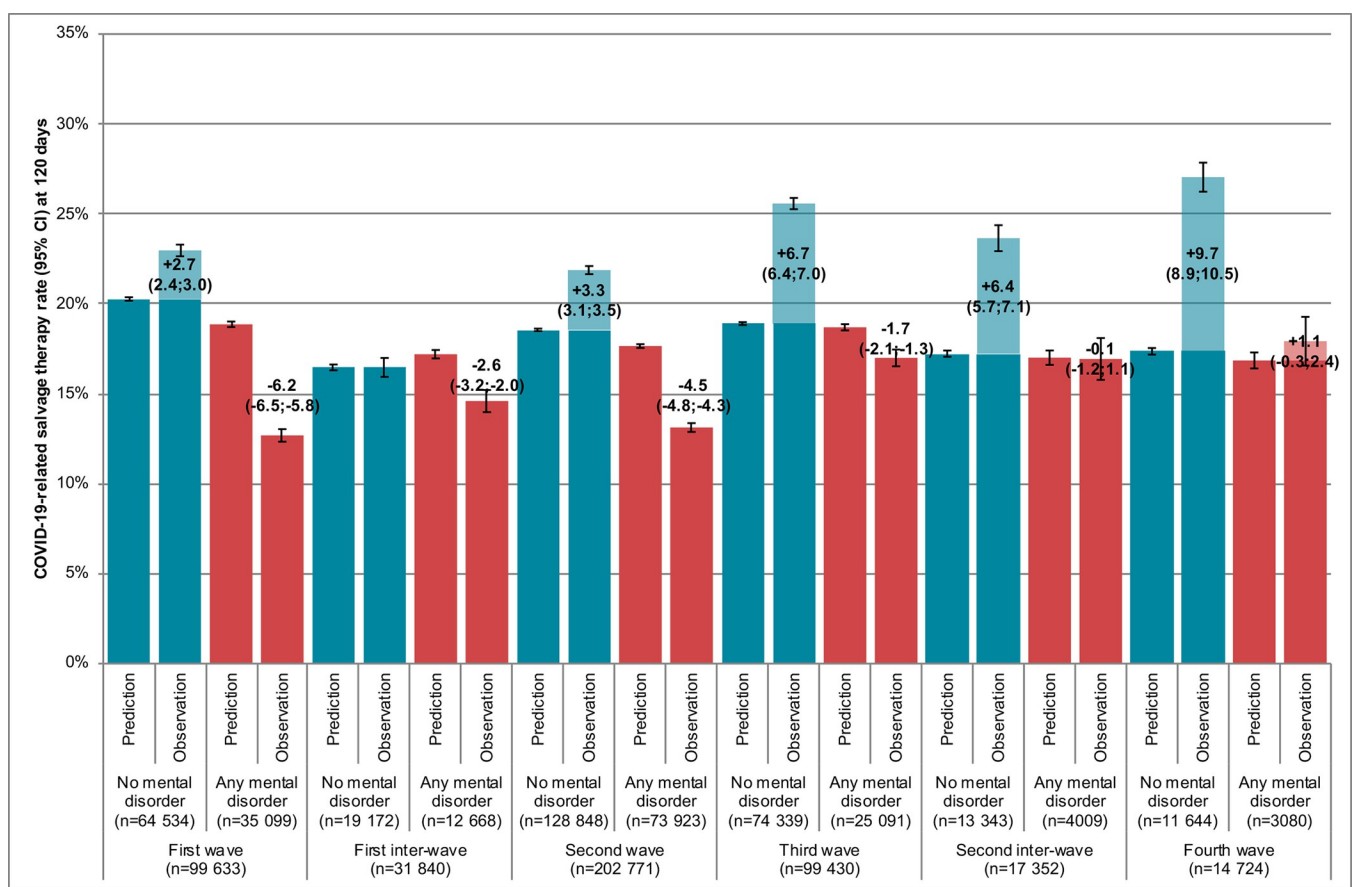

**Fig 3. Associations of pandemic periods and preexisting mental disorders with salvage therapy rate among inpatients with symptomatic COVID-19 in France (*n* = 465,750).**

increased COVID-19-related mortality. This result remained stable after controlling for pandemic periods over 18 months, sex, age, deprivation, lifestyle factors, somatic comorbidities, and other confounders. COVID-19 caseload surges in hospitals were associated with higher COVID-19-related mortality, although patients with mental disorders were disproportionately affected compared to others. In particular, the lower association of mental disorders with salvage therapy in all pandemic periods suggests that life-saving measures at French hospitals were disproportionately denied to patients with mental disorders in this exceptional context.

Our findings confirm the overall association between preexisting mental disorders and COVID-19-related mortality [5,6]. However, previous results were highly heterogeneous and inconclusive at hospital or for several categories of mental disorders [5,6]. In this nationally exhaustive sample of inpatients with symptomatic COVID-19, we found that all categories of mental disorders, except opioid use disorders [32], were independently associated with increased COVID-19-related mortality.

Several mechanisms have been hypothesised to explain the association of preexisting mental disorders with COVID-19-related mortality including barriers to care, social and lifestyle factors, higher rates of somatic comorbidities, and biological processes [5,6]. While the French health-care system provides universal health coverage [33] and hospital care for COVID-19 is free of charge, our study findings were fully adjusted on social factors (including deprivation), lifestyle factors (including tobacco smoking and obesity), and multiple severe somatic

comorbidities. Therefore, previous factors that were expectedly more frequently recorded among patients with preexisting mental disorders cannot explain the association. In addition, we found that almost all categories of mental disorders were independently associated with COVID-19-related mortality, suggesting that biological processes mainly hypothesised for psychotic disorders cannot explain the association.

Our study findings support that the association of preexisting mental disorders with COVID-19-related mortality was indirectly due to COVID-19 caseload surges that impacted triage decisions for life-saving measures at hospital. The hospital bed capacity in acute care has been continuously decreasing in OECD countries (e.g., from 4.1 beds per 1,000 inhabitants in 2000 to 3.0 beds in 2019 in France) [34]. In response to the first and second/third COVID-19 waves, the French government ordered 2 national lockdowns, along with a sharp reduction of regular admissions at hospital of non-COVID-19 patients. Nevertheless, about half a million inpatients were discharged with symptomatic COVID-19 over 18 months (i.e., 0.9% of the adult population in mainland France on January 1, 2020), and we found that hospital responsiveness was limited with excess mortality risks due to caseload surges of +5.0% (95% CI, 4.7 to 5.2) in 311,880 (67.0%) inpatients without mental disorders and +9.3% (95% CI, 8.9 to 9.8) in 153,870 (33.0%) inpatients with mental disorders.

Our case study of salvage therapy suggests that triage decisions for life-saving measures at hospital were disproportionately taken to maximise health benefits at the expense of COVID-19 patients with mental disorders: salvage therapy rates during caseload surges were significantly higher than expected in patients without mental disorders (+4.2% [95% CI, 3.8 to 4.5]) and lower in patients with mental disorders (−4.1% [95% CI, −4.4; −3.7]). In the COVID-19 crisis situation, French and international medical guidelines recommended prioritisation criteria to withhold or withdraw life-saving measures in vulnerable elderly patients with preexisting severe somatic conditions [35–37]. As expected, we found that numerous patients aged 80 years and older (34.3%), or recorded with dementia (14.5%), cerebrovascular disease (14.0%), or solid tumours (14.8%) had lower odds to accessing salvage therapy and poorer short-term prognosis, all other things being equal. In contrast, medical guidelines recommended that risk factors identified early on for severe COVID-19 (i.e., acute hospital admission) should not be considered to set priorities as they were not associated per se with COVID-19-related mortality in intensive-care units [35]. Our study findings suggest that these risk factors may, in fact, have been overused to triage patients into intensive-care units as patients recorded with these risk factors (25.2% with obesity in particular) had higher odds to accessing salvage therapy despite having better short-term prognosis, all other things being equal.

Our study findings suggest that preexisting mental disorders were generally not considered as risk factors for severe COVID-19 but rather as additional severe comorbidities, with seemingly lower priority given to them for life-saving measures and, ceteris paribus, poorer short-term prognosis. Alternatively, our study findings may reveal possible stigma and discrimination towards mental health or less support from family, carers, and friends in medical decisions [12]. Exceptions to this were found for rare patients with opioid use disorders (0.7%) or Down syndrome (0.2%), who may have benefited from early concerns in the COVID-19 pandemic [38,39] with higher or equal odds to accessing salvage therapy, respectively. Otherwise, the vast majority of patients with mental disorders only returned to equal odds to accessing salvage therapy after mid-May 2021 in relation to the mass COVID-19 vaccination campaign and decreasing caseload surges at hospital [20].

The main limitation of our study points to the assessment of COVID-19-related mortality. We assessed mortality within 120 days (22.3%) of the first date of recoded COVID-19 diagnosis at hospital to account for deaths following initial admissions in acute care with long length of stay [18], transfers to post-acute palliative care, or readmissions to acute care. Reassuringly,

consistent results were found when mortality was assessed at 28 days (17.2%) or first acute hospital discharge (19.2%). In particular, we found that excess mortality risks due to COVID-19 caseload surges in hospitals were about the same at 28 and 120 days in patients without mental disorders (about 5.0%) and patients with preexisting mental disorders (about 9.5%), pointing to an early cause in COVID-19 hospital management to explain difference between patient groups. We also assumed that deaths occurring within 120 days were related to COVID-19 in symptomatic COVID-19 inpatients. This study as well as other large cohort studies evidenced that "COVID-19-related deaths" occurred mostly in vulnerable elderly patients with severe somatic comorbidities [2–4] and actual counts may be overestimated [1]. However, consistent counts were reported for hospitals on French medical cause-of-death certificates (102,717 [70.4%] of 145,967 COVID-19-related deaths in 2020 to 2021) [40]. In addition, 34,024 (23.3%) COVID-19-related deaths were reported for nursing homes [40] supporting the evidence that patients with preexisting mental disorders (mostly dementia in nursing homes) generally had lower access to life-saving measures including lower odds to accessing acute hospitals.

Another limitation relates to potential coding bias of medical information recorded in hospital claims data. Regarding COVID-19 case definition, we found consistent results when selecting inpatients with COVID-19 respiratory symptoms (79.7%) or admitted for symptomatic COVID-19 at hospital (84.9%). Regarding other covariates, we relied on all discharge information recorded in all French hospitals over the last 9 years and we do not believe that the bias is large as 83.8% inpatients had at least 1 hospital admission before COVID-19. In agreement with previous reports [2–4], we found independent associations of older age, male sex, deprivation, and all severe somatic comorbidities except AIDS with higher mortality risk. Except for diabetes mellitus, we found that early-identified risk factors for severe COVID-19 (i.e., acute hospital admission) were not associated with higher mortality risk in a fully adjusted model. Previous results were highly heterogeneous in meta-analyses, and, similar to the case of mental disorders, may be explained by insufficient adjustment [23]. We considered various categories of mental disorder given their common stigmatisation [12], higher risk for premature death [10], and frequent inter-relationships [15] that were controlled for in this study. In particular, Down syndrome was included as a category of mental disorders in accordance with previous systematic reviews and meta-analyses [5,6] and frequent co-recording of mental retardation level (F70-F79) in the ICD-10 chapter of "mental and behavioural disorders" (F00-F99). Finally, we aimed at limiting residual confounding by assessing the timing of COVID-19 in the hospital trajectory of usually vulnerable elderly patients and found that a symptomatic SARS-CoV-2 infection occurring after hospital admission (15.1%) was independently associated with poorer short-term prognosis.

In conclusion, our study findings cast light on and question the triage system for salvage therapy with respect to COVID-19 patients with preexisting mental disorders in French acute hospitals. Future investigations should not only find out the reason for lower access for this patient group, but also how to set right such access in French hospitals. In addition, it would be interesting and important to see whether these results could be replicated in other countries.

## Supporting information

**S1 Text. Supporting information.** Appendix A. Coding dictionary. Appendix B. RECORD statement1 –checklist of items, extended from the STROBE statement, for observational studies using routinely-collected health data. Table A. Characteristics of inpatients with symptomatic COVID-19 by pandemic period (*n* = 465,750). Table B. 120-day mortality risk of inpatients with symptomatic COVID-19 by salvage therapy triage, univariate analyses (*n* = 465,750). Table C. 120-day mortality risk of inpatients with symptomatic COVID-19 by

salvage therapy triage, simultaneous probit multivariate model ($n = 465,750$). Table D. Controlled direct effects of preexisting mental disorders on 120-day mortality risk of inpatients with symptomatic COVID-19, causal mediation analyses ($n = 465,750$). Table E. Characteristics of inpatients with symptomatic COVID-19 by preexisting mental disorders ($n = 465,750$). Table F. Characteristics of inpatients with symptomatic COVID-19 by age category and preexisting mental disorders ($n = 465,750$). Table G. 120-day mortality and salvage therapy risks by category of preexisting mental disorders ($n = 465,750$). Fig A. Associations of pandemic periods and preexisting mental disorders with 120-day mortality risk among inpatients with symptomatic COVID-19 aged 18–64 years ($n = 164,591$). Fig B. Associations of pandemic periods and preexisting mental disorders with 120-day mortality risk among inpatients with symptomatic COVID-19 aged 65 years and above ($n = 301,159$). Fig C. Associations of pandemic periods and preexisting mental disorders with salvage therapy rate among inpatients with symptomatic COVID-19 aged 18–64 years ($n = 164,591$). Fig D. Associations of pandemic periods and preexisting mental disorders with salvage therapy rate among inpatients with symptomatic COVID-19 aged 65 years and above ($n = 301,159$). Fig E. Associations of pandemic periods and preexisting mental disorders with 120-day mortality risk among inpatients with COVID-19-related respiratory symptoms ($n = 371,016$). Fig F. Associations of pandemic periods and preexisting mental disorders on salvage therapy rate among inpatients with COVID-19-related respiratory symptoms ($n = 371,016$). Fig G. Associations of pandemic periods and preexisting mental disorders with 120-day mortality risk among inpatients admitted for symptomatic COVID-19 ($n = 395,323$). Fig H. Associations of pandemic periods and preexisting mental disorders on salvage therapy rate among inpatients admitted for symptomatic COVID-19 ($n = 395,323$). Fig I. Associations of pandemic periods and preexisting mental disorders with intensive-care unit admission rate among inpatients with symptomatic COVID-19 ($n = 465,750$). Fig J. Associations of pandemic periods and preexisting mental disorders with 28-day mortality risk among inpatients with symptomatic COVID-19 ($n = 465,750$). Fig K. Associations of pandemic periods and preexisting mental disorders with mortality risk at first acute hospital discharge among inpatients with symptomatic COVID-19 ($n = 465,750$). (DOCX)

## Author Contributions

**Conceptualization:** Michaël Schwarzinger.

**Data curation:** Michaël Schwarzinger.

**Formal analysis:** Michaël Schwarzinger.

**Methodology:** Michaël Schwarzinger, Stéphane Luchini.

**Software:** Michaël Schwarzinger.

**Validation:** Michaël Schwarzinger, Vincent Mallet, Jürgen Rehm.

**Writing – original draft:** Michaël Schwarzinger, Jürgen Rehm.

**Writing – review & editing:** Michaël Schwarzinger, Stéphane Luchini, Miriam Teschl, François Alla, Vincent Mallet.

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
