## [Editor Report · Decision Letter 0]

14 Jul 2022

Dear Dr Schwarzinger, 

Thank you for submitting your manuscript entitled "Mental disorders and COVID-19-related life-saving measures and mortality: a nationwide, retrospective hospital cohort study over an 18-month period in France" for consideration by PLOS Medicine.

Your manuscript has now been evaluated by the PLOS Medicine editorial staff and I am writing to let you know that we would like to send your submission out for external peer review.

Your manuscript is currently under consideration as part of the Special Issue on the COVID-19 pandemic and global mental health. The deadline for the Special Issue is being extended to December 15 2022, with anticipated publication in Q1 2023 (subject to change dependent on submission volume). We intend to publish all papers accepted for the Special Issue simultaneously.

Given that this extension was announced after you submitted your manuscript for consideration, we appreciate that you may no longer wish for your manuscript to considered specifically for the Special Issue. If this is the case, or if you have any other questions, please feel free to contact me (cdavidson@plos.org) and this can be discussed.

Before we can send your manuscript to reviewers, we need you to complete your submission by providing the metadata that is required for full assessment. To this end, please login to Editorial Manager where you will find the paper in the 'Submissions Needing Revisions' folder on your homepage. Please click 'Revise Submission' from the Action Links and complete all additional questions in the submission questionnaire.

Please re-submit your manuscript within two working days, i.e. by Jul 18 2022 11:59PM.

Kind regards,

Callam Davidson

Associate Editor

PLOS Medicine

---

## [Decision Letter · Decision Letter 1]

17 Aug 2022

Dear Dr. Schwarzinger,

Thank you very much for submitting your manuscript "Mental disorders and COVID-19-related life-saving measures and mortality: a nationwide, retrospective hospital cohort study over an 18-month period in France" (PMEDICINE-D-22-02340R1) for consideration at PLOS Medicine. 

Your paper was evaluated by an associate editor and discussed among all the editors here. It was also discussed with an academic editor with relevant expertise, and sent to independent reviewers, including a statistical reviewer. The reviews are appended at the bottom of this email and any accompanying reviewer attachments can be seen via the link below:

[LINK]

In light of these reviews, I am afraid that we will not be able to accept the manuscript for publication in the journal in its current form, but we would like to consider a revised version that addresses the reviewers' and editors' comments. Obviously we cannot make any decision about publication until we have seen the revised manuscript and your response, and we plan to seek re-review by one or more of the reviewers. 

We hope to receive your revised manuscript by Sep 07 2022 11:59PM. Please email us (plosmedicine@plos.org) if you have any questions or concerns.

We look forward to receiving your revised manuscript. 

Sincerely,

Callam Davidson, 

Associate Editor

PLOS Medicine

plosmedicine.org

Comment from the Academic Editor:

I do not see analyses which attempt to draw a causal inference between access to salvage therapy (which would have occurred during hospitalization and, always, before death) and mortality. This, in my view, diminishes the potential impact of the analyses. That said, the data is all there and I am sure a statistician with the relevant expertise (for e.g. in path/mediation analysis) could carry out analysis to examine the proportion of the excess mortality risk which could be attributed to lower access to salvage therapy. This, to my mind, would make this paper stand out from all others I know on the excess mortality risk for persons with mental disorders.

Please revise your title to ‘Mental disorders, COVID-19-related life-saving measures and mortality in France: a nationwide cohort study’, or similar.

Line 31: ‘have shown’ rather than ‘showed’.

Abstract Background: The final sentence should clearly state the study question.

Abstract Methods and Findings:

* Please include the years during which the study took place.

Line 53: ‘odds of’ rather than ‘odds in’.

Please ensure you address reviewer comments pertaining to the assessment of mental disorders (e.g., reviewer #3, comment #4) as this issue has the potential to introduce bias in the findings.

Please provide the rationale behind your decision to categorise Down’s syndrome and other learning disabilities as mental disorders. As noted by reviewer #1, it may be inappropriate to group genetic disorders associated with both mental and physical disabilities alongside other mental disorders. 

Please place citations before punctuation. 

Introduction: Please address past research and explain the need for and potential importance of your study. Indicate whether your study is novel and how you determined that. If there has been a systematic review of the evidence related to your study (or you have conducted one), please refer to and reference that review and indicate whether it supports the need for your study.

Please define salvage therapy in your Introduction.

Please conclude the Introduction with a clear description of the study question or hypothesis.

As noted by reviewer #3 (comment #3), the method used to attribute COVID-19 mortality requires justification as the editors felt the current approach could result in misclassification bias. 

Line 102: As noted by reviewer #1, this statement requires clarification, as your study does rely on data from human participants. 

Throughout (including Tables and Figures): Please report p values as P<0.001 rather than P<0.0001. 

Lines 191-193: Your study is observational, therefore please remove language that may imply causality (e.g., ‘effects of’ and ‘detrimental’). Refer instead to associations. 

Line 198: Related to the above, please reword ‘exacerbated’, to avoid implying a causal relationship.

Please define the dashed lines in the key for Figure 1. 

The data labels in Figures 2 and 3 are difficult to read, please enlarge the text slightly. 

Please define the error bars in Figures 2 and 3 in the figure legend or title. 

Please remove the COI information from References 1, 5, 8, 16, 17, and 20 (please check for others). 

Did your study have a prospective protocol or analysis plan? Please state this (either way) early in the Methods section.

Comments from the reviewers:

Reviewer #1: Thanks for the opportunity to review your manuscript. My role is as a statistical reviewer, so my review concentrates on the study design, data, and analysis that are presented. I have put general questions first, followed by queries relevant to a specific section of the manuscript (with a page/line reference).

The study uses routinely collected hospital data from France in a population of hospitalised patients with COVID-19, and assesses risk factors for mortality (with 120 days follow-up) and use of salvage therapy (respiratory support and ICU admission). The key focus is on differences in patients with and without mental disorders. This is estimate with multivariable logistic regression models, and the results extended to a gap analysis (where the outcome is predicted without the mental disorder variable) and the difference between predicted and actual is compared. 

There is some discussion about the limitation of measuring exposures from hospital records, one thing I think that needs to be mentioned is that some of the covariates/risk factors are ones that vary temporally and by classifying patients with any record of these (e.g. depression) as having that exposure, it assumes that the patient is still affected at the time of the COVID-19 admission. This might be best approached by shifting some of the language, e.g. 'patients with recorded history of depression'. 

This is getting out of my area of expertise, but I was not certain about the classification of Down's syndrome as purely a 'mental disorder'. This is usually classified primarily as a genetic disorder that commonly presents with both mental and physical disabilities, rather than a purely mental disorder.

P3, L94. I went to look at the information about the method of record linkage, and the website was in French (alas, French was not a study option for me in high school). With an extra sentence or two of detail would you be able to specify the record linkage method? E.g. was it a statistical linkage key? What were the identifiers used? Was it encrypted? 

P4, L102. Just wondering about the terminology, as the data is specifically from human participants. Is the law about research where human participants are not directly contacted? 

P4, L110. Was mortality recorded in the hospital records or taken from record linkage with national mortality records? Is there any way to check if most mortality was capture from hospital records, or is it possible that a significant amount of mortality outside of hospitals after separation was not captured?

P4, L124. To clarify, if someone had one of the disorders that define the main exposure in the previous 8 years they were assumed to still have the disorder at the time of the COVID-19 admission? This seems reasonable for some of the disorders (e.g. dementia) but some of these disorders (e.g. depression, alcohol) might not be present at the time of admission of interest, i.e. it is assumed that there is no variation in the disorder over time.

P5. L141. Very pleased to see that you have selected a key set of variables and not done a variable selection process - with the entire French population you have plenty of sample size to use all the covariates of interest. What criteria was used to select the particular covariates from the multitude of potential ones available in the hospital data? 

P5, L144. This is an interesting approach that reminds me of a standardised mortality ratio. Is it possible for the full details of this procedure to be include either with reference to a publication with the details in it or with the addition of the material into an appendix? This is important for the peer review process. 

I also couldn't see in the manuscript where p-values from the t-test procedure where applied. Is this done across the entire dataset for actual vs. predicted outcomes, with a binary outcome? If binary then a McNemar's test would probably be appropriate than a test intended for a continuous variable. Given the sample size though I think the CI is much more useful for inference.

Table S1/S2 (and similar tables throughout the appendix). Given the large sample size and descriptive nature of the tables I think the p-value is superfluous. 

Reviewer #2: The manuscript submitted by Dr. Schwarzinger et al. aims to analyze the association between mental disorders and COVID-19 related life-saving measures and mortality. The paper is very clearly written and of great quality. Authors use an impressive and exhaustive hospital dataset which enables them to go further previous studies. Their main finding is that COVID-19 patients with mental disorders had lower odds in access to ICU during the pandemic in France and had higher mortality rates. 

Major comments: 

- Statistic procedures: the authors state that 'we considered that patients without any mental disorder during the first inter-wave period had the best prognostic outcome'. What if this assumption is broken? It would be good to have a sensitivity analysis on this, and discuss the choice of the reference period, especially given that all results are based on this analysis.

- Similarly, the rate of testing in France changed dramatically over the course of the pandemic, and tests were still not fully available in the first inter-wave period (see raw data here: https://www.ecdc.europa.eu/en/publications-data/covid-19-testing), with large delays between test and results in July-August 2020. Would test availability have an impact on the results?

Minor comments:

- Line 67: it would be good to specify which 'specific categories' were inconclusive in other papers.

Reviewer #3: This is a review of the manuscript "Mental disorders and COVID-19-related life-saving measures and mortality: a nationwide, retrospective hospital cohort study over an 18-month period in France" submitted for publication in PLOS Medecine. This manuscript addresses a timely, important and debated research question, i.e. whether mental disorders are independently associated with COVID-19-related mortality and access to salvage therapy. However, the current submission may suffer from several methodological and statistical issues that weaken its conclusion, but most issues could be relatively easily addressed. Below are several suggestions that I hope the authors will find useful.

1/ Introduction: "Mental disorders also seem to increase the risk of serious COVID-19 outcomes, especially mortality, as two systematic reviews and meta-analyses have suggested". To the reviewer's knowledge, this point is debated. Two recent studies taking into account many confounders, and in particular medical comorbidities, found that respectively psychiatric disorders (doi: 10.1038/s41380-021-01393-7) and major depression (doi: 10.1371/journal.pone.0255427) could be significantly associated with reduced mortality, possibly due to the potentially protective role of certain psychotropic medications (PMID: 34608263). Because of this debate, it would be important to detail and shortly discuss both the measures of associations and the prevalence rates of mental disorders in those meta-analyses. Indeed, the prevalence rates of major psychiatric disorders in Barcella et al. (PMID: 33894064) and Vai et al. (PMID: 34274033) meta-analytic studies were very low (3.6% and 3.1%, respectively), which may suggest that a small proportion of people with psychiatric disorders, those with important medical comorbidities, have increased risk of COVID-19-related mortality.

Methods and statistics:

- 2/ Population selection: Why excluding all patients discharged alive after day-case admission or recorded with asymptomatic SARS-CoV-2 infection. If there is a significant difference in the prevalence of psychiatric disorders in this subsample compared to the one used in the analysis, it may introduce a selection bias. In addition, for patients hospitalized (even for several hours) for several health conditions, it may be difficult to distinguish the specific contribution of SARS-CoV-2 infection, for which the symptoms and its impact were still not completely known. Finally, patients with COVID-19 and a day-case admission could have received oxygen therapy. Therefore, I strongly suggest including in the main analysis all these patients with a COVID-19 diagnosis record to avoid such potential selection bias. Sensitivity analyses excluding these patients can be presented as supplementary analysis.

- 3/ Mortality attributable to COVID-19 versus to other causes? Since the authors are interested in COVID-19-related deaths, the 120-day mortality may not be appropriate. According to the London Imperial College (https://www.imperial.ac.uk/), the mean delay between infection onset and death is 16.0 (SD= 8.2) days. Using 120-day mortality, it is highly probable that a substantial proportion of deaths is not attributable to COVID-19. Given that people with versus without mental disorders are more likely to die (form cardiovascular disorders, suicide, etc…) (doi:10.1192/bjp.173.1.11), the results may wrongly attribute this excess mortality to COVID-19. My recommendation would be to use 28-day mortality in the main analysis, as usually done in RCTs, to address this point, and modify accordingly Table 1.

- 4/ Mental disorders assessment: The 9-year timeframe for assessing mental disorders is very discussable, as for example people who had 1 major depressive episode 7 years ago but no depressive symptoms for the last 5 years cannot be considered as having a diagnosis of psychiatric disorder. Given the extremely high prevalence of psychiatric disorders reported in this study (33%), it suggests that most people "with mental disorders" have a past history but not a current diagnosis of psychiatric disorders, limiting the interpretability of the findings. A more conservative 1-year history of psychiatric disorders before AND NOT during the hospitalization with/for COVID-19 would clarify the link between preexisting "active" psychiatric disorders and COVID-19-related mortality. Importantly, mental disorders should not be assessed based on the diagnosis record of the hospitalization with/for COVID-19, since COVID-19 is significantly associated with increased incidence of neurological or psychiatric diagnoses (PMID: 33836148), and these patients with mental disorders related/consecutive to COVID-9 are more likely to have had severe COVID-19, and consequently increased risk of death. To avoid this bias, I suggest to distinguish 3 groups of patients: i) with a past-year diagnosis of mental disorder (excluding the hospitalization with COVID-19), (ii) no 9-year history of mental disorders, and (iii) incident diagnosis of psychiatric disorders (during the hospitalization and not in the past 9 years). This analysis would help distinguish the true effect of pre-existing current psychiatric disorders from that of secondary psychiatric disorders, such as frequently seen in ICUs, which may simply reflect severe COVID-19.

- 5/ Adjusting for comorbidity: although the Charlson Comorbidity Index is described in the methods, this variable is not reported in the result section or in the tables. Because medical comorbidity is a central confounder, I think it would be important to add, beyond all individual diagnoses, the total number of comorbidities (categorized in quintiles for example) to examine the specificity of associations between individual disorders, such as psychiatric disorders, and mortality (doi: 10.1038/s41380-021-01393-7). 

- 6/ Adjusting for BMI: because patients with mental disorders are more likely to have altered BMI and because CDC data indicates a J-shaped (nonlinear) relationship between continuous BMI and disease severity (PMID: 33705371), BMI should be studied with a greater number of degrees of freedom, as recommended by the CDC: i.e., underweight (<18.5 kg/m2), healthy weight (18.5-24.9 kg/m2 [reference]), overweight (25-29.9 kg/m2), obesity (30-34.9 kg/m2), obesity II (35-39.9 kg/m2), obesity III (35-39.9 kg/m2), obesity IV (40-44.9 kg/m2), and obesity V (≥45 kg/m2). 

- 7/ Additional potentially important confounders of the association between mental disorders and mortality could be (i) the total number of medications, known to increase mortality (PMID: 32909235), (ii) the lower vaccination rates in patients with mental disorders (PMID: 35753318), and (iii) a greater disease severity when hospitalized due to a greater mean delay between infection onset and hospitalization. If available, these variables should be included in the models; if not, this may constitute important limitations that should be acknowledged. 

- 8/ "To limit residual confounding from multiple other severe somatic conditions, we assessed the delay between the latest acute hospital discharge for any reason other than pregnancy and psychiatry and first COVID-19 diagnosis record": it should be better explained how this reduces the residual confounding; in addition, not taking into account hospital discharge for psychiatric disorder may introduce a bias since the reason of hospitalization in Psychiatry is not so rarely for both psychiatric and non-psychiatric reasons (e.g., dementia with behavioral problems due to non-psychiatric reasons). 

- 9/ Tables 1 and 2 should also provide descriptive statistics for people alive. 

- 10/ Table 1. The AOR adjusted for "any mental disorder" should be provided as it corresponds to the main result of the study. The difference for any mental disorder (33% versus 30.5%) does not look very substantial and may not even be significant. 

- 11/ Details about the main multivariable logistic regression models for "any mental disorder" should be provided in supplementary material: list of variables included in the models (all covariates should be included), number of degrees of freedom, quality checks with residuals and collinearity diagnostics. 

- 12/ Given the apparently substantial and robust association between mental disorders and lower access to salvage therapy, it would be very interesting to perform a supplementary analysis testing the association between any mental disorder and mortality, while adjusting for all potential confounders listed in Table 1 as well as salvage therapy rate. A potentially non-significant or a reversed association would be interesting to discuss as it may help reconcile all prior findings highlighted in 1/

- 13/ I would shortly discuss the potential impact of multiple testing on the results.

Reviewer #4: The manuscript has great public health value and well presented. Just I have the following few comments. 

1. Why authors used categorical data instead of continuous scores? 

2. On page 7 and 8, the authors stated that "Our case study of salvage therapy suggests that triage decisions for life-saving measures at hospital were 259 disproportionately taken to maximize health benefits at the expense of COVID-19 patients with mental disorders: salvage therapy rates due to caseload surges were significantly higher than expected in 261 patients without mental disorders (+4.2% [95% CI, 3.8-4.5]) and lower in patients with mental disorders 262 (-4.1% [95% CI, -4.4;-3.7]". What can be the possible explanations for this finding? can stigma against people with mental illness also be one possible explanation? 

3. Title of the tables and graphs should give full information.

[LINK]

---

## [Decision Letter · Decision Letter 2]

6 Oct 2022

Dear Dr. Schwarzinger,

Thank you very much for re-submitting your manuscript "Mental disorders, COVID-19-related life-saving measures and mortality in France: a nationwide cohort study" (PMEDICINE-D-22-02340R2) for review by PLOS Medicine.

I have discussed the paper with my colleagues and the academic editor and it was also seen again by two reviewers. I am pleased to say that provided the remaining editorial and production issues are dealt with we are planning to accept the paper for publication in the journal.

[LINK]

We look forward to receiving the revised manuscript by Oct 13 2022 11:59PM.   

Sincerely,

Callam Davidson, 

Associate Editor 

PLOS Medicine

plosmedicine.org

Requests from Editors:

Please consider expanding slightly on ‘French law’ in the Data Availability Statement (e.g., ‘French data protection law’, or similar, assuming this is accurate).

Thank you for your reply to my previous comment regarding categorisation of Down’s syndrome and other learning disabilities. The editorial team appreciate your rationale here and would ask that you include slightly more detail in the Discussion to clarify this point (the current wording at line 190 can be expanded using the content in your rebuttal letter as a basis). 

Line 72: Please update ‘was not explored’ to ‘has not been fully explored’, or similar.

Line 198: ‘Following the editor’s comment’ - please update the wording here to ‘In response to the peer review process’, or similar. 

Line 205: ‘Based on previous findings’ – while I appreciate that this wording was included in response to a reviewer comment, I feel it lacks clarity and creates the impression that you’re referring to previously published work. Please consider rephrasing to make it clear that your analysis here was data-driven (as opposed to pre-planned) and based on findings in the present study. 

Line 216: ‘Previous approach’ – please update to ‘This approach’, or similar.

Line 235: ‘decreased in 2021 in relation to COVID-19 vaccination uptake’ – although this factor may account for the lower median age observed in the latter waves, the current data do not support a causal statement – please temper as appropriate (e.g., ‘potentially’).

S2 Table: Please confirm whether P-value for opioid disorders is correct (should it be P<0.001 or 0.006?).

S6 Table: Please report as P<0.001 rather than P<0.0001.

Line 293: Please update to ‘admitted for symptomatic COVID-19’.

Comments from Reviewers:

Reviewer #1: Thanks for the revised manuscript and responses to my initial queries. The additional information on the linkage key was useful for my understanding about the completion of the data, thank you. The additions to the manuscript are helpful in clarifying some of my initial queries (legal basis for use of data) and fairly describe what the data represents (e.g. pre-existing conditions). The explanation about the method used for the 'excess mortality' and the causal mediation model is good, it was a good suggestion from the academic editor to include the CMM. 

I am happy with the explanation of the inclusion of Down's syndrome based on the previous meta-analyses. As this is outside my area of expertise I can't offer an informed opinion as to whether the labelling of this particular condition needs some modification. 

The only minor amendment I suggest is the p-values in Table 1 and Table S1. I agree with the authors that appending p-values to a table like this could be considered a tradition in medical research journals, but it is also a tradition whose time has come: https://doi.org/10.1080/00031305.2016.1154108 . The small p-values reflect a large sample size and all the relevant information can be gained from the actual summary data presented.

[LINK]

---

## [Editor Report · Decision Letter 3]

25 Oct 2022

Dear Dr Schwarzinger, 

On behalf of my colleagues and the Academic Editor, Professor Vikram Patel, I am pleased to inform you that we have agreed to publish your manuscript "Mental disorders, COVID-19-related life-saving measures and mortality in France: a nationwide cohort study" (PMEDICINE-D-22-02340R3) in PLOS Medicine.

PRESS

PLOS frequently collaborates with press offices. If your institution or institutions have a press office, please notify them about your upcoming paper at this point, to enable them to help maximise its impact. If the press office is planning to promote your findings, PLOS would be grateful if they could coordinate with medicinepress@plos.org. As this manuscript is to be published as part of the upcoming Special Issue on the pandemic and global mental health, it will be opted out of the early version process. If you would like to discuss this further or if you have any further questions or concerns, please reach out directly (cdavidson@plos.org).

We also ask that you take this opportunity to read the PLOS Embargo Policy regarding the discussion, promotion and media coverage of work that is yet to be published by PLOS. As your manuscript is not yet published, it is bound by the conditions of our Embargo Policy until publication as part of the Special Issue. Please be aware that this policy is in place both to ensure that any press coverage of your article is fully substantiated and to provide a direct link between such coverage and the published work. For full details of our Embargo Policy, please visit http://www.plos.org/about/media-inquiries/embargo-policy/.

Sincerely, 

Callam Davidson 

Associate Editor 

PLOS Medicine